# Discovery and characterization of a terpene biosynthetic pathway featuring a norbornene-forming Diels-Alderase

Zuodong Sun [1], Cooper S. Jamieson [2], Masao Ohashi[1], K. N. Houk [1,2 ✉] & Yi Tang [1,2 ✉]

Pericyclases, enzymes that catalyze pericyclic reactions, form an expanding family of enzymes that have biocatalytic utility. Despite the increasing number of pericyclases discovered, the Diels-Alder cyclization between a cyclopentadiene and an olefinic dienophile to form norbornene, which is among the best-studied cycloadditions in synthetic chemistry, has surprisingly no enzymatic counterpart to date. Here we report the discovery of a pathway featuring a norbornene synthase SdnG for the biosynthesis of sordaricin-the terpene precursor of antifungal natural product sordarin. Full reconstitution of sordaricin biosynthesis reveals a concise oxidative strategy used by Nature to transform an entirely hydrocarbon precursor into the highly functionalized substrate of SdnG for intramolecular Diels-Alder cycloaddition. SdnG generates the norbornene core of sordaricin and accelerates this reaction to suppress host-mediated redox modifications of the activated dienophile. Findings from this work expand the scopes of pericyclase-catalyzed reactions and P450-mediated terpene maturation.

[1] Department of Chemical and Biomolecular Engineering, University of California, Los Angeles, Los Angeles, CA 90095, USA. [2] Department of Chemistry and Biochemistry, University of California, Los Angeles, Los Angeles, CA 90095, USA. ✉email: houk@chem.ucla.edu; yitang@g.ucla.edu

One of the best-studied pericyclic reactions is the [4 + 2] cycloaddition between a cyclopentadiene and a substituted olefinic dienophile to form a bridged bicyclic norbornene (Fig. 1a). This Nobel prize winning reaction, first studied by Diels and Alder in their 1928 seminal publication[1], has become the prototype for cycloaddition and demonstrated many important features of Diels-Alder (DA) reactions such as stereoselectivity[2], the concerted mechanism[3,4], and various mechanisms of rate acceleration[5–7]. Surprisingly, cycloadditions involving a cyclopentadiene to form norbornene-containing compounds have not been found in biosynthesis. Instead, most reported biosynthetic DA reactions take place with a linear diene derived from unsaturated acyclic precursors[8–15] (Fig. 1b). Searching the natural product database for norbornene-containing metabolites yielded less than 200 documented structures[16], a vast majority of which are plant-derived adducts formed between two sesquiterpenes via proposed cycloadditions, such as the diguaianolide absinthin[17]. The fungal-derived sordarin (Fig. 1c) is the only family of microbial natural products that contains a norbornene core and is speculated to form via an intramolecular Diels-Alder (IMDA) reaction[18].

Sordarin (Fig. 1c) is a diterpene glycoside isolated from the fungus *Sordaria araneosa* Cain[19]. Sordarin and various derivatives are potent antifungal agents through the inhibition of fungal elongation factor 2[20,21]. Biosynthesis of different sordarins is proposed to diverge from norbornene-containing precursor

sordaricin (**1**, Fig. 1c), which has been coisolated with sordarins[22]. Because of its unique structure among microbial natural products, sordaricin has inspired numerous synthetic studies[23–26], with several centered around a biomimetic IMDA step between synthetically generated diene-dienophile pairs. However, the IMDA reactions in these synthetic studies were performed in organic solvent and required prolonged reaction time (3 days)[23,25]. In addition, preparation of the synthetic diene-dienophile pair required at least 15 steps with an overall yield less than 3%[23,25]. It is therefore of interest to understand how Nature biosynthesizes sordaricin. The putative biosynthetic gene cluster (BGC) of a highly decorated sordarin derivative, hypoxysordarin (*sdn*) (Fig. 1c) from *S. araneosa*, was reported by Kudo and coworkers (Fig. 1d)[27]. The *sdn* cluster is anchored by a diterpene synthase SdnA which was shown to cyclize geranylgeranyl diphosphate (GGPP) into the 5-8-5 tricyclic hydrocarbon cycloaraneosene, the putative precursor to sordaricin (Fig. 1c, Supplementary Fig. 1)[27,28]. Notwithstanding these findings, the biosynthetic strategy to generate the reactive species for cycloaddition and the nature of the IMDA reaction (enzymatic vs uncatalyzed) are unresolved.

In particular, it is intriguing how the hydrocarbon cycloaraneosene can be morphed to form the norbornene core in sordaricin. To the best of our knowledge, IMDA reaction has not been reported to take place during the maturation of a terpene natural product, although one example of cycloaddition between

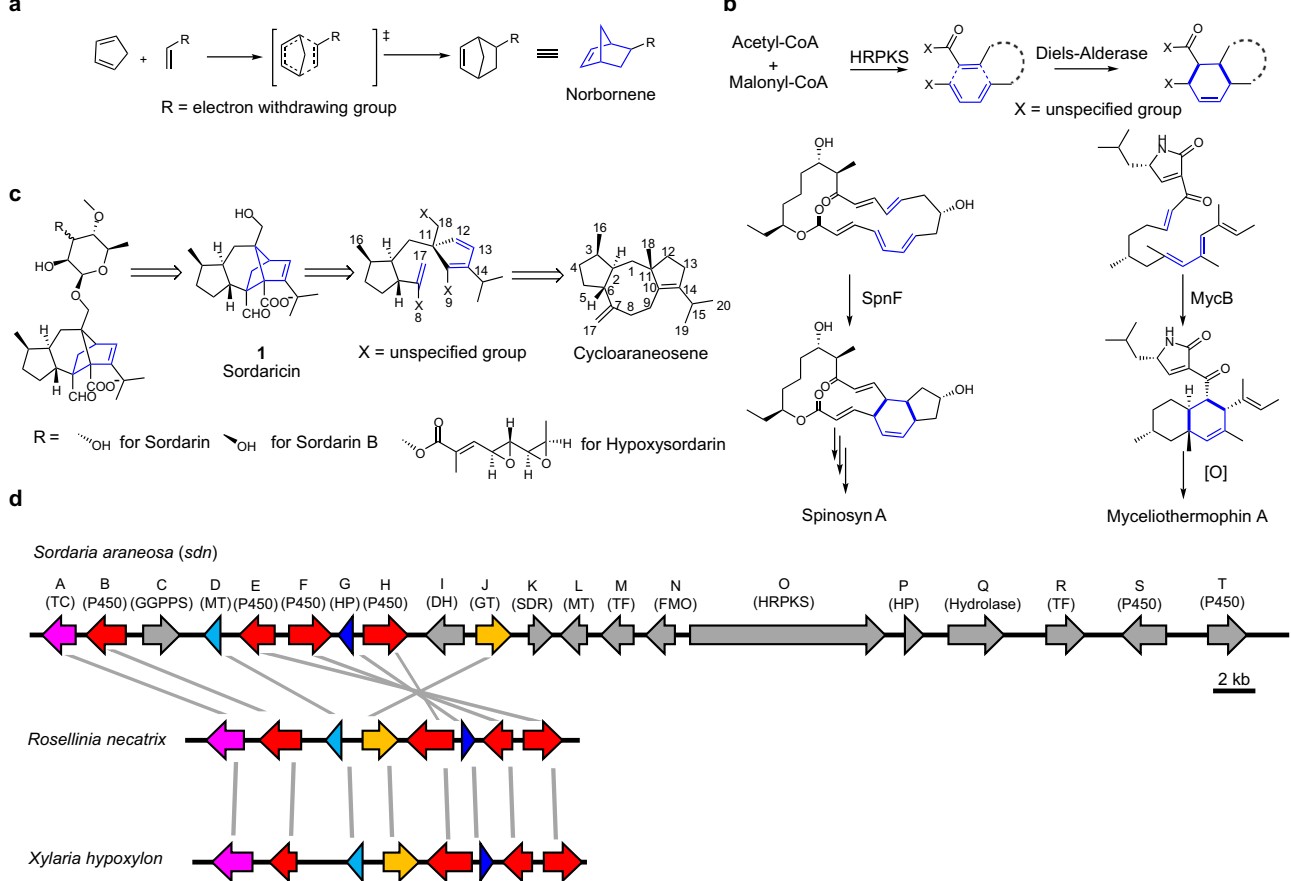

**Fig. 1 Biosynthesis of sordaricin is an example of the classical norbornene forming Diels-Alder reaction in Nature. a** DA reaction between a cyclopentadiene and a substituted olefinic dienophile yielding norbornene. **b** Well characterized IMDA reactions in biosynthesis of polyketides and polyketide-nonribosomal peptide hybrid natural products. HRPKS, highly reducing polyketide synthase. **c** Retro-biosynthetic scheme of sordarins. The numbering of cycloaraneosene follows Kudo, et al[27]. **d** BGCs for hypoxysordarin (*sdn*) and sordarin B (putative). Genes in *sdn* but absent from the sordarin B cluster are colored gray. TC, terpene cyclase; GGPPS, geranylgeranyl pyrophosphate synthase; MT, methyl transferase; HP, hypothetical protein; DH, dehydratase; GT, glycosyl transferase; SDR, short chain reductase; TF, transcription factor; FMO, flavin-dependent monooxygenase.

a sesquiterpene dienophile and a polyketide-derived quinone methide diene[29] has been documented[30]. Extensive modifications of the cycloaraneosene skeleton are expected to ready the molecule for norbornene formation: (1) unlike IMDA reactions observed for polyketide chains that are conformationally flexible to position dienes and dienophiles in spatial proximity (Fig. 1b)[8–15], the polycyclic terpene molecule is rigid, thereby requiring breaking one or more C-C bonds to afford rotational freedom; (2) desaturation of $sp^3$-$sp^3$ C-C bond(s) in the cyclized terpenes such as cycloaraneosene must take place to generate the diene moiety; and (3) the diene and especially the dienophile require activation to lower the transition state (TS) energy barrier for IMDA. Whereas dienophiles in polyketides that undergo IMDA are typically conjugated to electron withdrawing groups as a result of polyketide synthase programing, the hydrocarbon scaffold of a terpene molecule must be strategically oxidized prior to the pericyclic reaction.

Here, we report the complete reconstitution of sordaricin biosynthesis from cycloaraneosene and the chemical logic that setups an IMDA reaction to form the norbornene structure. A pericyclase that accelerates the IMDA cycloaddition and attenuates competing shunt product formation is discovered and characterized. The findings in this work represent an example of a pericyclic reaction involved in building terpene structural complexity.

## Results

**Identification of genes likely involved in sordaricin biosynthesis.** The previously identified hypoxysordarin BGC (*sdn*, GenBank accession: LC079035.1, https://www.ncbi.nlm.nih.gov/nuccore/LC079035.1) contains twenty genes (SdnA-SdnT), a majority of which are expected to encode enzymes required in the maturation of sordaricin to the final product hypoxysodarin[27] (Fig. 1d, Supplementary Fig. 1). Given that genes encoding enzymes for sordaricin biosynthesis must be conserved in the BGCs of all sordarin analogs, we searched the sequenced fungal genome data for more *sdn*-like clusters for comparative analysis. Using SdnA as a query, a more compact cluster was found to be conserved in *Rosellinia necatrix* and *Xylaria hypoxylon* (Fig. 1d). This cluster contains eight genes, all of which are present in the larger *sdn* cluster. In addition to SdnA, the homologous *sdn* enzymes include four P450 oxygenases (SdnB, SdnE, SdnF and SdnH) and a hypothetic protein (SdnG). A glycosyltransferase (SdnJ) and methyltransferase (SdnD) are also conserved, although these two enzymes are not expected to participate in sordaricin formation based on the putative functional annotation. We predict these more compact pathways are responsible for sordarin B biosynthesis (Fig. 1c, Supplementary Table 1), during which SdnJ glycosylates the sordaricin core with rhamnose[22], while SdnD methylates the 4-hydroxy group of the transferred rhamnose. Therefore, we putatively assigned SdnA and the four P450s to be involved in sordaricin biosynthesis, with potential participation by the hypothetic protein SdnG. A flavin-dependent monooxygenase SdnN, which was proposed to play a central role in oxidative maturation of cycloaraeosene, is not conserved between *sdn* and the more compact clusters[27] (Fig. 1d, Supplementary Fig. 1).

**SdnB is a multifunctional P450 in the *sdn* pathway.** To reveal the chemical logic for transforming cycloaraeosene into sordaricin, we reconstituted the four *S. araneosa* P450 enzymes SdnB/E/F/H with SdnA in the heterologous host *Aspergillus nidulans* A1145 ΔEMΔST[31]. SdnC, a GGPP synthase presents in the *sdn* cluster but not in the more compact clusters, was included to increase GGPP flux in the host. Expression of SdnA and SdnC led to the production of cycloaraneosene[27] (Supplementary Fig. 2).

We then coexpressed SdnA and SdnC with each of the four P450s (SdnB, E, F, and H) individually. While SdnE, SdnF, and SdnH did not transform cycloaraneosene to new metabolites, coexpression of SdnB yielded two new metabolites **2** (25 mg/L) and **3** (3.5 mg/L) (Fig. 2a, Supplementary Fig. 3a). Structural characterization via nuclear magnetic resonance (NMR) and high-resolution mass spectrometry (HRMS) established **2** as (8 *R*, 9 *S*)-cycloaraneosene-8,9-diol, which is derived from two successive hydroxylation of cycloaraneosene (Fig. 2b, Supplementary Notes, Supplementary Table 4, Supplementary Figs. 4, 14–19). The stereochemistries of the diol **2** were determined by NOESY based on the reported stereochemistry for cycloaraneosene[27,28]. The monohydroxylated cycloaraneosene-8-ol was previously isolated from *S. araneosa* and is likely an intermediate leading to **2**[27] (Supplementary Fig. 1).

Structure elucidation of **3** showed an unexpected shunt product and revealed an additional role of SdnB in the pathway (Fig. 2b, Supplementary Notes, Supplementary Table 5, Supplementary Figs. 4, 20–24). We proposed that **3** is likely formed via oxidative cleavage of the C-8, C-9 diol in **2** by SdnB (Fig. 2b), which was confirmed through direct feeding of **2** to *A. nidulans* expressing only SdnB (Fig. 3a). Compound **2** is stable under the same feeding conditions when an empty plasmid control was used. Since the conversion of **2** to **3** is a net redox-neutral process, the product of SdnB oxidation is likely dialdehyde **4** instead of **3**. In the absence of downstream enzymes, the C-7 acrolein moiety in **4** can be reduced by endogenous reductases in *A. nidulans* to afford alcohol **3** as a cellular detoxification mechanism to remove the unsaturated aldehyde[32,33]. We chemically synthesized **4** by selectively oxidizing the allylic alcohol in **3** to the corresponding unsaturated aldehyde via activated $MnO_2$[34] (Supplementary Notes, Supplementary Table 6, Supplementary Figs. 4, 25–29). Consistent with our hypothesis, **4** was readily converted to **3** when fed to *A. nidulans* expressing only empty plasmids (Fig. 3a). Overall, our results suggest that SdnB both acts as a canonical monooxygenase and also a thwarted oxygenase, an oxygenase that consumes oxygen to generate strong enzymatic oxidant but does not result in formal incorporation of oxygen atom into the product[35] (Supplementary Fig. 5). The diol cleavage activity of SdnB enables rotation of the C-6-C-7 bond and thereby freed the C-7-C-17 double bond which is the proposed dienophile for the IMDA reaction (Fig. 2b).

**SdnH is a desaturase that generates the cyclopentadiene.** Each of the remaining P450 enzymes (SdnE, SdnF, or SdnH) was then coexpressed with SdnA-C-B to identify the next biosynthetic step. Only the coexpression of SdnH led to two new metabolites **5** (7 mg/L, Supplementary Notes, Supplementary Table 7, Supplementary Figs. 4, 30–34) and **6** (4.5 mg/L, Supplementary Notes, Supplementary Table 8, Supplementary Figs. 4, 35–39) (Fig. 2, Supplementary Fig. 3b). Both compounds are oxidatively cleaved at C-8 and C-9 as in **3**, and both contain the cyclopentadiene functionality. Compound **5** is reduced at C-8 as in **3**, whereas **6** is further oxidized from **5** at C-9 to a carboxylate. Similar to the formation of **3** from **4**, **5** is likely a redox shunt product derived from cyclopentadiene-dialdehyde **7**, which should be the product of sequential actions of SdnB and SdnH starting from cycloaraneosene (and **2**). Auto-oxidation or host oxidases may subsequently convert **5** to **6**. Our results suggest SdnH, another thwarted oxygenase, catalyzes the desaturation of C-12-C-13 of the cyclopentene ring present in cycloaraneosene to generate the cyclopentadiene (Supplementary Fig. 5).

Two routes can be proposed for the formation of **7** from the diol **2**, either via **4** (SdnB followed by SdnH) or via **8** (SdnH followed by SdnB). Comparing the relative titers of shunt products in the heterologous strains suggests that the latter route is in play. *A.*

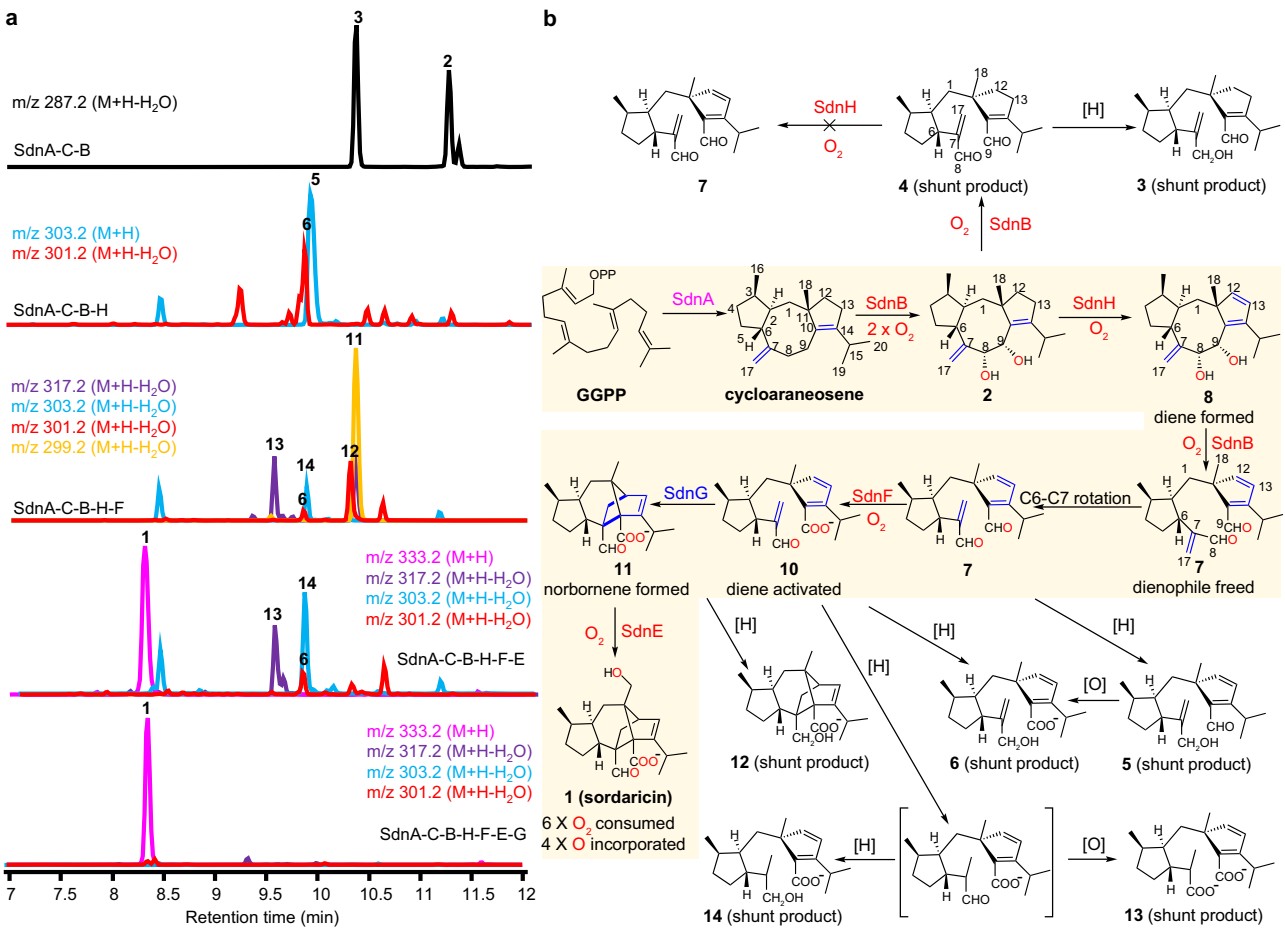

**Fig. 2 Sordaricin biosynthesis involves a concise yet well-programed chemical logic. a** Stepwise heterologous reconstitution of genes involved in sordaricin biosynthesis. The chromatograms are extracted from mass spectra of the base peak for each compound. **b** Complete biosynthetic pathway for sordaricin. The main pathway is highlighted with a shaded background. All numbered compounds are structurally characterized via NMR and HRMS. The compound in bracket is proposed but not observed.

*nidulans* expressing SdnA-C-B accumulates 25 mg/L of diol **2** but only 3.5 mg/L of **3**, suggesting **2** is a suboptimal substrate of SdnB. In contrast, **2** was greatly diminished in *A. nidulans* expressing SdnA-C-B-H, with **5** and **6** being the predominant products. This suggests that the desaturation activity of SdnH is sandwiched between the diol-forming and oxidative cleavage activities of SdnB in the pathway, with **8** as a biosynthetic intermediate. To assay the activity of SdnH directly, we performed biotransformation of **2** in both *A. nidulans* and in *Saccharomyces cerevisiae* expressing SdnH. In both strains, **2** was readily transformed into **8** (Fig. 3b). Isolation and characterization of **8** from yeast confirmed the compound is the cyclopentadiene-containing diol (40% isolation yield from **2**, Supplementary Notes, Supplementary Table 10, Supplementary Figs. 4, 45–50). The stereochemistries of **8** were determined by NOESY based on the reported stereochemistry for cycloaraneosene[27,28]. Further feeding of **8** to *A. nidulans* expressing SdnB led to near complete conversion to **5** (Fig. 3c), supporting the proposal that **8** is the true on-pathway intermediate. Lastly, feeding **4** to *A. nidulans* expressing SdnH did not give cyclopentadiene-containing products, suggesting that SdnH only recognizes the intact 5-8-5 ring system in **2** (Fig. 3d) and desaturation must occur before C-8-C-9 cleavage.

**SdnF oxidation activates the diene for IMDA.** The cyclopentadiene-containing **7** does not undergo IMDA, as evidenced in the metabolic profile of SdnA-C-B-H expression strain.

As a result, **7** is subjected to cellular redox modifications to shunt products **5** and **6**. These compounds are also unable to form norbornene due to electronically mismatched substitutions on diene and dienophile pair. To examine the reactivity of **7**, we chemically synthesized **7** by activated $MnO_2$ oxidation of the allylic alcohol in **5** (Fig. 4a, Supplementary Notes, Supplementary Table 9, Supplementary Figs. 4, 40–44). During synthesis in dichloromethane, we observed and characterized the norbornene-dialdehyde **9** as a minor product (Supplementary Notes, Supplementary Table 11, Supplementary Figs. 4, 51–56). We monitored the uncatalyzed cyclization of **7** by following the disappearance of 304 nm absorption from the cyclopentadiene moiety (Supplementary Fig. 4). In a pH 7.4 HEPES buffer, **7** cyclizes with a $k_{uncat}$ of 0.0018 $min^{-1}$, which corresponds to a half-life of 390 min (Fig. 4c inset). Hence, additional modification to the diene/dienophile pair to align the HOMO/LUMO energies is necessary to form sordaricin. The presence of the C-9 carboxylate group in sordaricin hints oxidation of the C-9 aldehyde in **7** to **10** is the logical next step.

To support this proposed C-9 oxidative activation, we performed density functional theory (DFT) calculations to understand the transition state energy barrier for IMDA starting from **7** (C-8, C-9 dialdehyde, TS-1), **10** (C-8 aldehyde, C-9 carboxylate, TS-2), and **5** (C-8 alcohol, C-9 aldehyde, TS-3) (Fig. 4e). The calculated $\Delta G^{\ddagger}_{uncat}$ for cyclization of **10** is 20.3 kcal/mol, which is ~7 kcal/mol lower than that for dialdehyde **7**

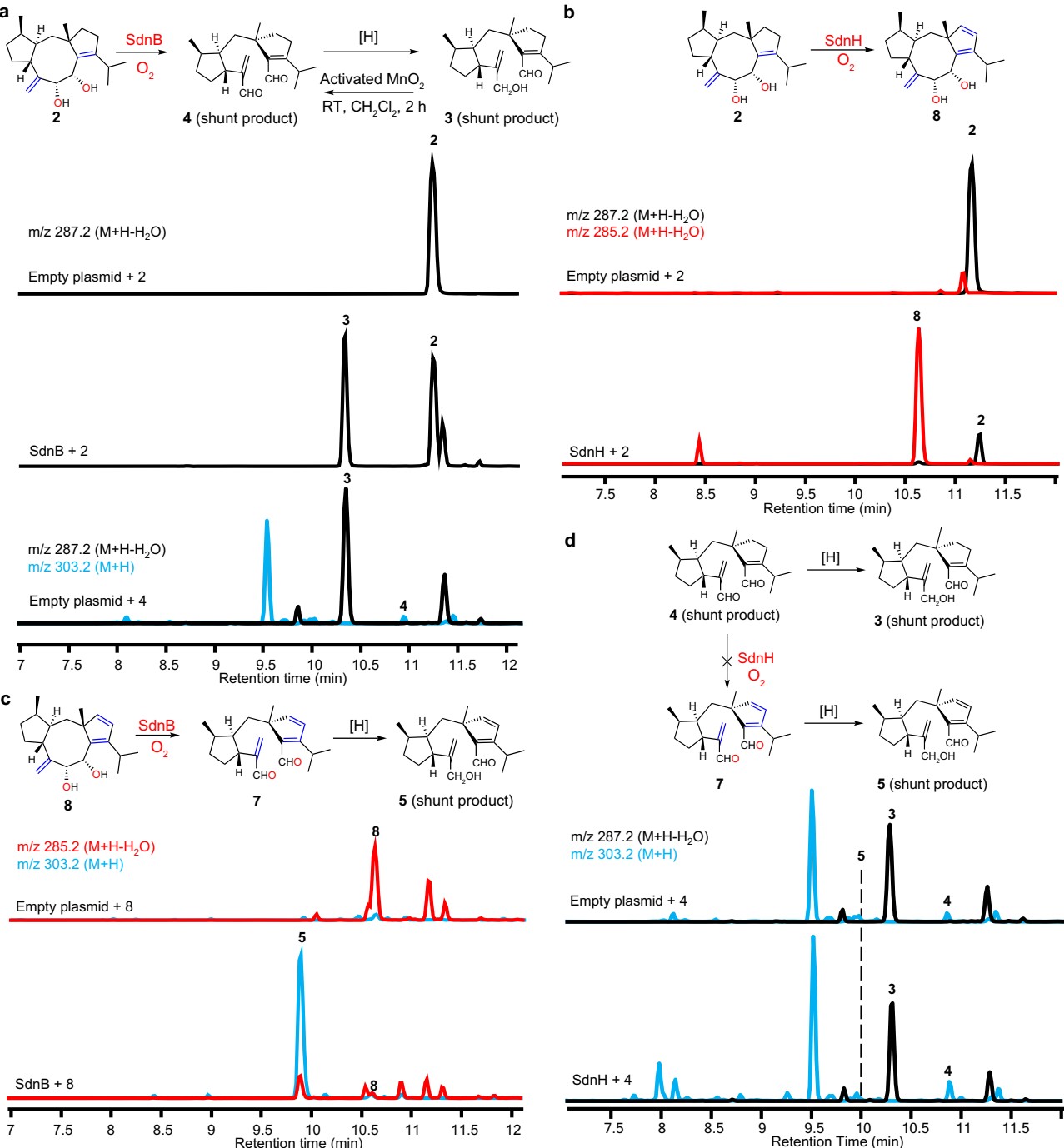

**Fig. 3 Biotransformation probing the function of SdnB and SdnH. a** Biotransformation of **2** by *A.nidulans* expressing SdnB. **b** Biotransformation of **2** by *A.nidulans* expressing SdnH. **c** Biotransformation of **8** by *A.nidulans* expressing SdnB. **d** Biotransformation of **4** by *A.nidulans* expressing SdnH. Compound **4** cannot be desaturated by SdnH to form **7** (and then to **5**) but instead was reduced to **3** by endogenous reductases in *A. nidulans*. The chromatograms in all cases are extracted from mass spectra of the base peak for each compound.

(27.6 kcal/mol) and represents significant rate enhancement upon C-9 oxidation to the carboxylate. The sluggish cyclization of dialdehyde **7** can be rationalized since both diene and dienophile in **7** are electron deficient. Oxidation of C-9 aldehyde to carboxylate eliminates the electron withdrawing aldehyde and accelerates cyclization of **10**. The calculation also confirmed a much higher activation barrier (33.9 kcal/mol) for the DA cycloaddition of **5**, consistent with the removal of the C-8 electron-withdrawing group conjugated to the dienophile.

One of the two remaining P450 enzymes is a likely candidate for the oxidation of **7** to **10**. Indeed, upon coexpression of SdnF with SdnA-C-B-H in *A. nidulans*, two new norbornene-containing metabolites **11** (1 mg /L) and **12** (1 mg/L) were formed with the concomitant disappearance of **5** and **6** (Fig. 2, Supplementary Fig. 3c). Compound **11** contains the cyclized norbornene ring and is one additional C-18 hydroxylation step from **1** (Supplementary Notes, Supplementary Table 13, Supplementary Figs. 4, 62–66). Compound **12** is a shunt product derived

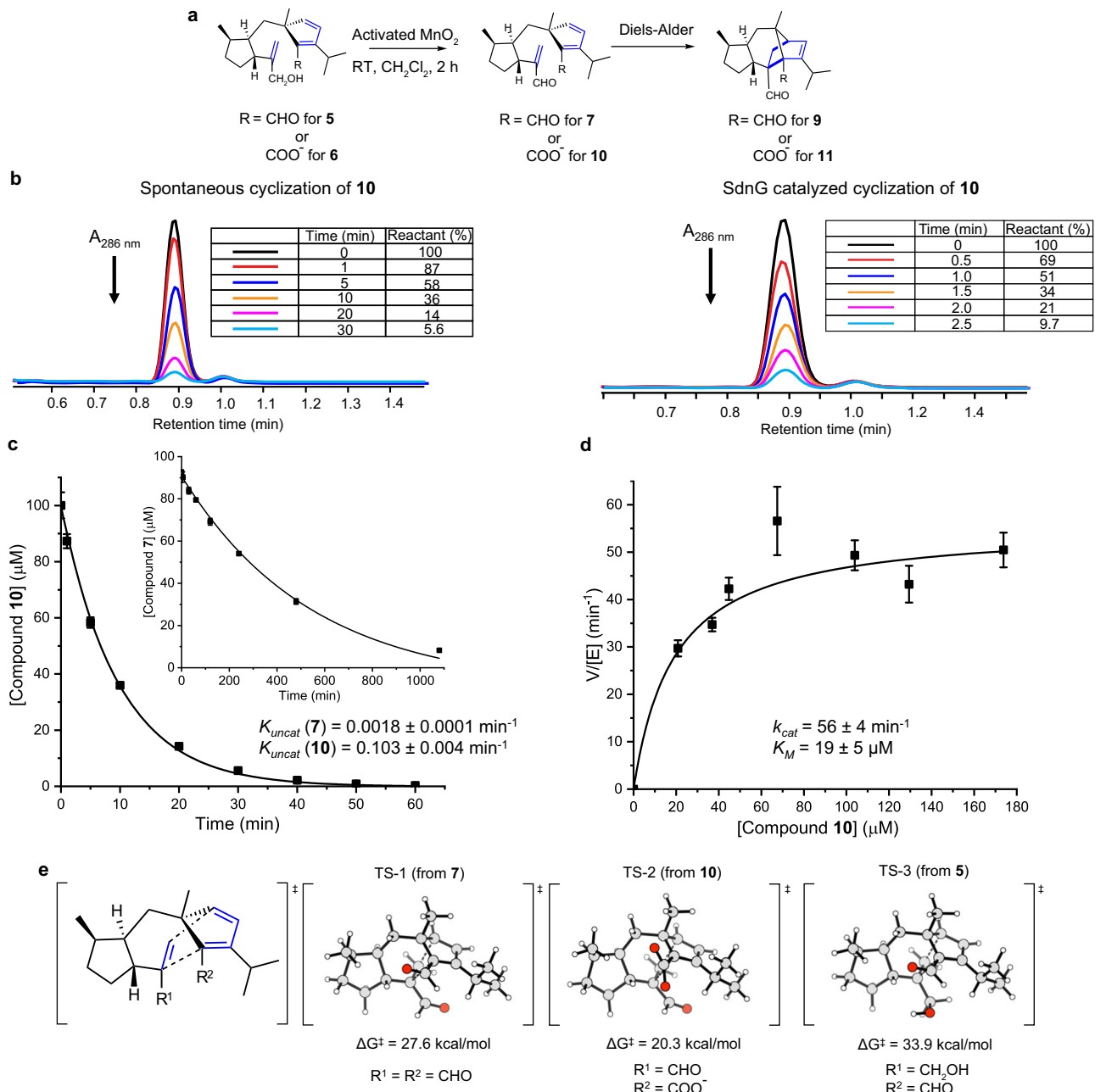

**Fig. 4 In vitro characterization of non-enzymatic and enzymatic norbornene formation in sordaricin biosynthesis. a** Overall synthetic scheme for compound **7** and **10**. **b** Cyclization of compound **10** in the absence (left) and presence (right) of SdnG in a 50 mM HEPES buffer, pH 7.4. Decrease at $A_{286\,nm}$ (disappearance of the cyclopentadiene chromophore) was used to monitor the cyclization. **c** First order kinetics of non-enzymatic cyclization of compounds **10** and **7** (inset). The data points and error bars represent the mean and standard deviation of $n = 3$ independent measures respectively. Errors associated with the kinetic parameters were obtained from fitting. Source data are provided as a Source Data file. **d** Michalis-Menten kinetics of SdnG with compound **10** as substrate. The data points and error bars represent the mean and standard deviation of $n = 3$ independent measures respectively. Errors associated with the kinetic parameters were obtained from fitting. Source data are provided as a Source Data file. **e** DFT calculations of IMDA transition states in water for dialdehyde **7** (TS-1), and carboxylate-aldehyde **10** (TS-2), and aldehyde-alcohol **5** (TS-3).

from **11** via reduction of C-8 aldehyde to alcohol (Supplementary Notes, Supplementary Table 14, Supplementary Figs. 4, 67–71). When chemically prepared **7** was fed to *A. nidulans* expressing SdnF alone, efficient conversion to **11** was observed (Fig. 5a). To assay the oxidation activities of SdnF separately from cyclization of **10**, **5** was supplemented to the SdnF expression strain and a control strain expressing only empty plasmids. While the control strain is able to convert ~30% of **5** to the carboxylate **6** likely via auto-oxidation or host oxidases, the SdnF expression strain led to

near complete conversion of **5** to **6** (Fig. 5b), demonstrating that SdnF is able to selectively oxidize the C-8 aldehyde to carboxylate.

To measure the rate of IMDA cyclization after C-8 oxidation, we chemically synthesized **10** by oxidizing **6** with activated $MnO_2$ and followed its cyclization by HPLC (Supplementary Notes, Supplementary Table 12, Supplementary Figs. 4, 57–61). In a pH 7.4 HEPES buffer, **10** cyclizes to form **11** with a $k_{uncat}$ of 0.103 $min^{-1}$, which corresponds to a half-life of 6.7 min for **10** (Fig. 4b, c). Compared to dialdehyde **7**, the carboxylate-aldehyde

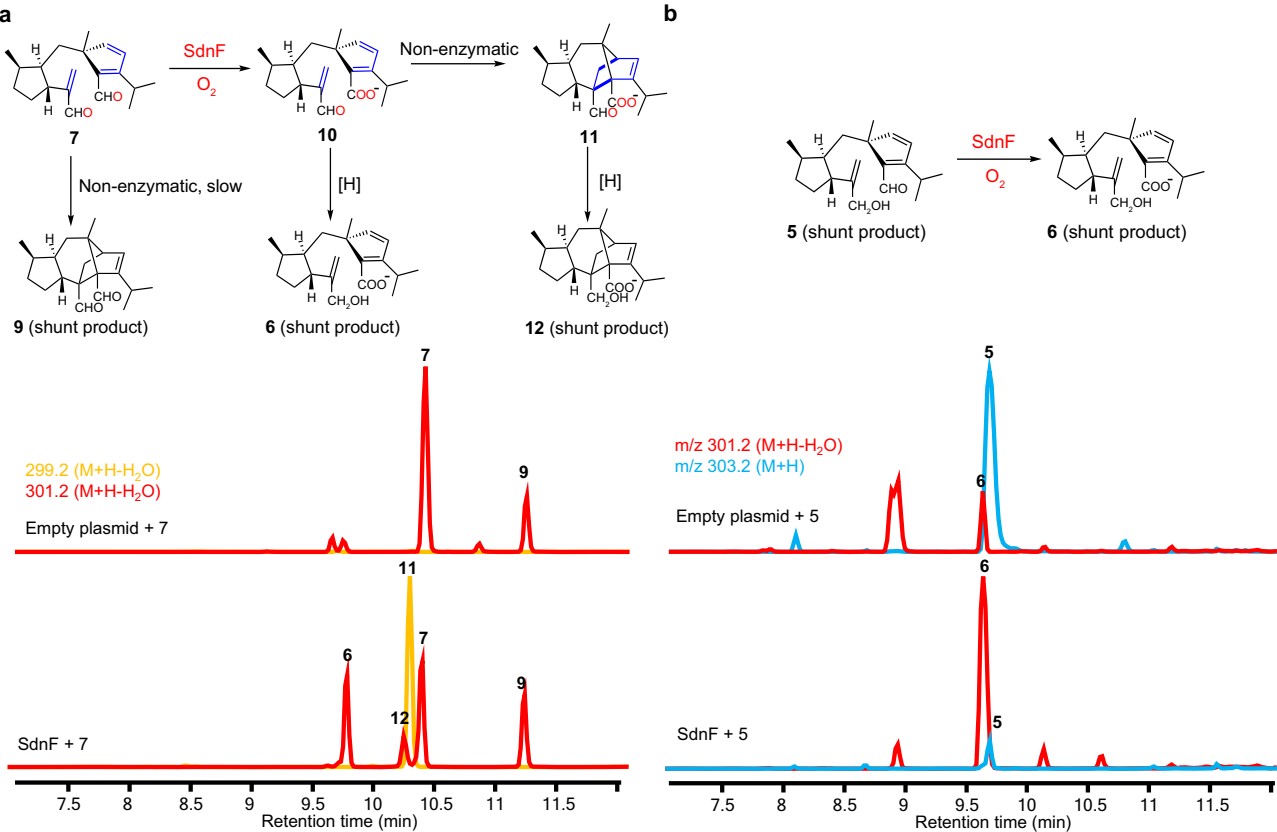

**Fig. 5 Biotransformation probing the function of SdnF. a** Biotransformation of **7** by *A.nidulans* expressing SdnF. **b** Biotransformation of **5** by *A.nidulans* expressing SdnF. The chromatograms in all cases are extracted from mass spectra of the base peak for each compound.

**10** is 57-fold more active towards IMDA cyclization. Therefore, SdnF activates the cyclopentadiene for norbornene formation.

**Complete reconstitution of sordaricin biosynthesis**. To complete sordaricin biosynthesis, we introduced the remaining P450 SdnE in *A. nidulans* expressing SdnA-C-B-H-F. The resulting host indeed biosynthesized sordaricin **1** (20 mg/L) via hydroxylation of the C-18 methyl in **11** (Fig. 2). The $^1$H and $^{13}$C NMR spectra and specific rotation of **1** match with those reported for sordaricin[25] (Supplementary Notes, Supplementary Table 3, Supplementary Figs. 4, 9–13). The pathway was also successfully reconstituted in *S. cerevisiae* RC01 with a titer of 2 mg/L (Supplementary Fig. 6). Notwithstanding the complete reconstitution of sordaricin pathway, we observed co-accumulation of previously isolated shunt product **6**, as well as two new shunt metabolites **13** (2.3 mg/L) and **14** (1.5 mg/L) (Fig. 2a). The same three shunt products were also present together with **5** in the strain that coexpressed SdnA-C-B-H-F (Fig. 2a). Scaled up cultures led to isolation and characterization of **13** (Supplementary Notes, Supplementary Table 15, Supplementary Figs. 4, 72–76) and **14** (Supplementary Notes, Supplementary Table 16, Supplementary Figs. 4, 77–81) as uncyclized shunts derived from reduction of the dienophile in **10**. Furthermore, the C-8 aldehyde in **13** is oxidized to carboxylate, while reduced to alcohol in **14**. The molar ratio of combined shunt products to sordaricin **6** is 1 to 4, suggesting that at least 20% of **10** is diverted to shunt pathways that outcompete the IMDA reaction. This is not unexpected since the measured $k_{uncat}$ of 0.103 min$^{-1}$ for **10** is relative slow compared to those expected for endogenous enzyme-catalyzed redox modifications. This apparent biosynthetic inefficiency prompted us to examine if additional biosynthetic enzymes are needed to accelerate the IMDA reaction and prevent shunt product accumulation.

**SdnG is a norbornene synthase**. To test if the *sdn* uses a dedicated enzyme to further accelerate the IMDA reaction of **10** to **11**, the remaining uncharacterized enzyme SdnG conserved between the *sdn* and more compact clusters shown in Fig. 1d was expressed in the strains that produced **1**. The predicted open reading frame of SdnG encodes a 146 amino acid protein with no conserved domain. A BLAST search in NCBI genome database did not identify any homologs of SdnG outside *sdn*-like clusters (Supplementary Fig. 7). When coexpressed, a dramatic change in metabolite profile was observed in which **1** is now nearly the exclusive product, with the shunt products present only at trace levels (Fig. 2a). Similarly, when SdnG was coexpressed in strain that produced **11** (SdnA-C-B-H-F), shunt product titers were greatly attenuated (Fig. 6a). Suppression of shunt products suggests SdnG acts as a pericyclase, more specifically a Diels-Alderase, to accelerate the IMDA reaction and outpace cellular redox modifications. It is worth noting here that once the final molecule **1** is formed, we can only detect trace amount of reduction or oxidation of the C-8 and C-9 groups in *A. nidulans*, for reasons we do not fully understand.

To verify the role of SdnG, the protein was expressed and purified from *E. coli* and assayed directly in the presence of the putative substrate carboxylate-aldehyde **10** (Supplementary Fig. 8a). In the absence of SdnG, up to 90% **10** (100 μM) is cyclized within 30 min, whereas cyclization of 90% **10** was observed within 2.5 min upon addition of 1 μM SdnG (Fig. 4b). This rate acceleration requires SdnG since a generic globular protein such as bovine serum albumin (BSA) does not accelerate cyclization of **10** (Supplementary Fig. 8b). The steady-state

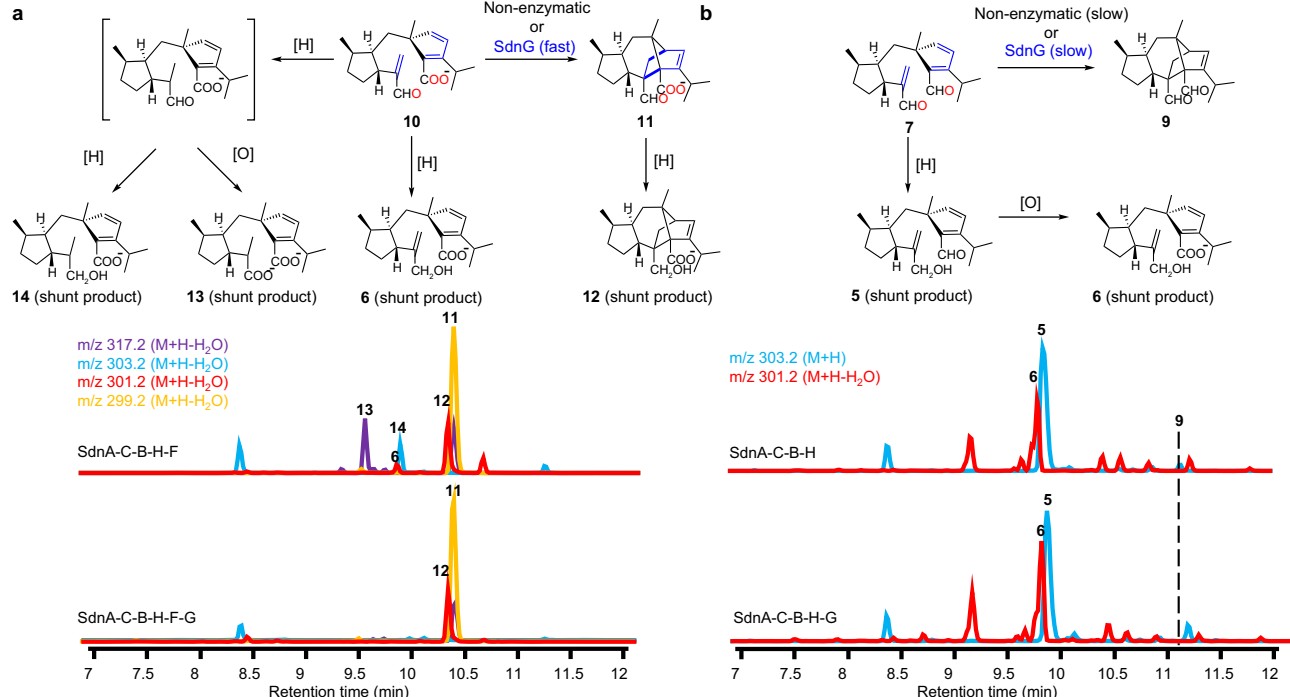

**Fig. 6 SdnG catalyzes efficient in vivo cyclization of 10 but not 7. a** Metabolic profiles of *A. nidulans* transformed with SdnA-C-B-H-F in the absence and presence of SdnG. **b** Metabolic profiles of *A. nidulans* transformed with SdnA-C-B-H in the absence and presence of SdnG. The cyclization of **7** is not fast enough to compete with shunt redox pathways (to form **5** and **6**) and only trace amounts of **9** is formed regardless the absence or presence of SdnG. The chromatograms in all cases are extracted from mass spectra of the base peak for each compound.

kinetics of SdnG was measured to give $K_M$ towards **10** of 19 μM and $k_{cat}$ of 56 min$^{-1}$ (Fig. 4d). The value of $k_{cat}$ is comparable to previously characterized pericyclases[9,11,13,15] and corresponds to a ~560-fold acceleration of the $k_{uncat}$. This rate acceleration is similar to that reported for SpnF, the first Diels-Alderase characterized[9]. SdnG is specific for the carboxylate-aldehyde **10**, as its pericyclase activity towards **10** is at least 100-fold higher than dialdehyde **7** in vitro (Supplementary Fig 8b). This is consistent with coexpression of SdnG with SdnA-C-B-H in *A. nidulans*, which did not lead to detectable amount of norbornene product **9** (Fig. 6b). Together, the results demonstrate SdnG as a dedicated Diels-Alderase catalyzing norbornene formation.

## Discussion

Our bioinformatics, heterologous reconstitution, and biotransformation analysis revealed the complete biosynthetic pathway for sordaricin. This pathway features a concise chemical logic to transform the linear primary metabolite GGPP to a highly functionalized norbornene scaffold via only one terpene cyclase and a four-P450 cascade that consumes six molecules of oxygen and incorporates four oxygen atoms into the final product (Fig. 2b). All four double bonds in GGPP are utilized at two distinct stages to construct the final product, with two forging the 5-8-5 core of the initial diterpene cycloaraneosene and the other two becoming critical components of a subsequent Diels-Alder cyclization. The terpene cyclase SdnA appears to be a unique enzyme since its product cycloaraneosene has only been associated with sordaricin biosynthesis. Despite structure similarity of its 5-8-5 core to other diterpenes such as fusicoccadiene[36], cyclooctat-9-en-7-ol[37], and ophiobolin M[38], cycloaraneosene has a distinguishing exocyclic olefin at C-7 which becomes the dienophile for the IMDA reaction (Fig. 1c).

Following SdnA, three P450s catalyze a series of intriguing reactions to overcome the intrinsic difficulty for a terpene scaffold to undergo IMDA reactions. The dihydroxylation of cycloaraneosene and the diol cleavage of **8** catalyzed by SdnB immediately rearrange the cycloaraneosene scaffold to free the C-7 dienophile from the rigid tricyclic scaffold. Our mechanism of dienophile release differs significantly from the previous proposal where the C-8-C-9 bond is to be cleaved via Baeyer-Villiger oxidation and the subsequent hydrolysis of resulting ester[27] (Supplementary Fig. 1). P450-catalyzed vicinal diol cleavage to yield fragmented aldehydes has been reported in biosynthesis of pregnenolone[39], biotin[40], and botrydial[41], but the SdnB reaction is particularly interesting since in this case the cleavage of one C-C bond readies the molecule to form two new C-C bonds via the later DA reaction. The idea of break a bond to make a bond, or molecular editing, has been increasingly employed in total synthesis of complex natural products as a strategy to reshuffle the molecular scaffold and generate new reactivity[42]. SdnB functions represent a biosynthetic example of generating the diene/dienophile pair via such carbon skeleton reconstruction.

The desaturation catalyzed by SdnH generates a key conjugated diene which otherwise are absent in a typical terpene. Previous study by Kudo, et al. suggested that the cyclopentadiene was generated by C-13 hydroxylation of **2** and the subsequent dehydration of the resulting cycloaraneosene triol via an unspecified enzyme[27] (Supplementary Fig. 1). Our results show that SdnH alone is sufficient for the desaturation. While P450 desaturases can perform both direct desaturation and hydroxylation, it is generally believed that the hydroxylated compounds are premature OH rebound products rather than intermediates of the desaturation activity[43]. The disruption of SdnB's hydroxylation and diol cleavage activities by SdnH is reminiscent of the multifunctional P450 TamI in tirandamycin biosynthesis[44]. Such an

unusual reaction sequence is likely programed to avoid premature generation of the reactive acrolein dienophile which is cytotoxic and leads to redox shunt products.

Besides freeing the dienophile, the diol cleavage catalyzed by SdnB serves another purpose towards the formation of sordaricin: installing conjugated aldehyde groups to both the dienophile and the cyclopentadiene. While the dienophile is sufficiently activated with an electron deficient aldehyde group, the HOMO/LUMO energy gap of the diene/dienophile pair is widened by a similarly electron deficient aldehyde group on the diene. Therefore, a third P450 SdnF is necessary to further activate the diene/dienophile pair for IMDA by converting the aldehyde conjugated to the diene to a less electron withdrawing carboxylate group. Such redox activation of diene/dienophile is analogous to the role played by the flavin-dependent enzyme Sol5 catalyzing a decalin-forming IMDA in solanapyrone biosynthesis where Sol5 is required to activate the dienophile by oxidizing an electron donating hydroxymethyl group conjugated to the dienophile to an electron withdrawing aldehyde[45,46]. Whereas P450 modification of terpene scaffolds are widely found in biosynthetic pathways, the collective roles played by SdnB, SdnH and SdnF to set up the IMDA is a rare catalytic strategy.

We also discovered a Diels-Alderase catalyzing norbornene formation. Although the IMDA reaction of **10** to **11** can take place uncatalyzed and stereoselectivity is substrate-controlled[25], the *sdn* pathway uses a dedicated pericyclase SdnG to accelerate the reaction as a means to suppress shunt product formation. It is not clear how endogenous redox metabolism towards reactive species such as the acrolein dienophile in the sordarin producers differs from that of *A. nidulans*, but in the heterologous host the redox modifications drain pathway intermediates towards uncyclized shunt products. SdnG contains no recognizable cofactor binding domain and does not dependent on any exogenous cofactors for activity, joining a group of cofactor-free pericyclases including SpnF[9], CghA[47], PyrI4[48], IccD[12], and the recently reported Tsn11[49]. However, SdnG shares no sequence homology with any characterized pericyclases and therefore likely represents yet an undescribed class of pericyclases. Further structural and functional characterizations of SdnG are currently underway to reveal mechanistic insights of such a norbornene synthase.

Norbornene ring formation from a cyclopentadiene and an olefinic dienophile has been a mainstay of mechanistic investigation and synthetic use of Diels-Alder chemistry for the past 80–90 years. The sordaricin biosynthetic pathway featuring SdnG represents an example of a biologically accelerated DA reaction to construct this classical bicyclic ring system. The pathway discovered here utilizes a concise yet well-programed chemical logic that is unmatched by chemical synthesis. The small size and the cofactor-free nature of the norbornene synthase SdnG sets up further mechanistic and biocatalytic exploration of pericyclases. Together, our finding adds to the ever-expanding reservoir of biological pericyclic reactions[12,50] and enriches Nature's toolbox for building terpene complexity. Lastly, synthetic sordarin analogs derived from sordaricin have shown great promise as leads for antifungal therapy[51–53]. Our results provide a direct and scalable route to sordaricin.

## Methods

**Bioinformatics**. BGCs containing cycloaraneosene synthase SdnA was identified by a BLAST search of SdnA homologs in NCBI genomic database (https://www.ncbi.nlm.nih.gov/genome/). Genomic scaffolds containing SdnA homologs were annotated by 2ndfind[54]. The NCBI Conserved Domain Search (https://www.ncbi.nlm.nih.gov/Structure/cdd/wrpsb.cgi) was used for conserved domain analysis.

**Genomic DNA extraction from *S. araneosene* and cDNA synthesis**. *S. araneosene* NRRL 3196 was obtained from the Agricultural Research Service Culture Collection (NRRL) and maintained on potato dextrose agar (Sigma). A 10 mL

liquid culture of *S. araneosene* NRRL 3196 in potato dextrose broth (Sigma) was shaken for 7 days at 28 °C and 250 rpm. The cell body was then collected for genomic DNA extraction with Quick-DNA™ Fungal/Bacterial Miniprep Kit (Zymo research) following the manufacturer's protocol.

For mRNA extraction and cDNA synthesis, a 10 mL liquid culture of *S. araneosene* in production medium (10% glucose, 1.5% polypeptone, 1.0% corn steep liquor, 0.5% yeast extract, 0.2% L-tryptophan, 0.5% $K_2HPO_4$, 0.4% $FeSO_4 \cdot 7H_2O$, 0.05% $CoSO_4$, 0.1% $MgSO_4 \cdot 7H_2O$) was shaken for 10 days at 28 °C and 250 rpm[27]. The cell body was then collected for mRNA extraction with RiboPure™ Yeast RNA Purification Kit (Thermofisher) following the manufacturer's protocol. First strand synthesis was performed with SuperScript™ III First-Strand Synthesis System (Invitrogen).

**General DNA manipulation techniques**. The DNA sequence of previously reported *sdn* cluster (GenBank accession: LC079035.1, https://www.ncbi.nlm.nih.gov/nuccore/LC079035.1) was used for all subsequent studies[27]. Plasmids for overexpression in *A. nidulans* were constructed as follows. The genes involved in sordaricin biosynthesis were amplified from genomic DNA of *S. araneosa* via polymerase chain reaction (PCR) and inserted into plasmids pYTU, pYTR, and pYTP via homologous recombination in *Saccharomyces cerevisiae* YJB77[55]. Empty vectors were linearized via restriction digestion with PacI and NheI or MluI (NEB). PCR was performed with Phusion (Thermofisher) or Q5 (NEB) high fidelity polymerases. Transformation of yeast was performed with Frozen-EZ Yeast Transformation II kit (Zymo research) following the manufacturer's instructions. Plasmids were then extracted from yeast with Zymoprep Yeast Plasmid Miniprep I (Zymo research) following the manufacturer's instructions and transformed into *E. coli* Top10 electrocompetent cells (Invitrogen) for propagation. Plasmids extracted from Top10 (Zyppy Plasmid Miniprep Kit, Zymo Research) were screened via Sanger sequencing (Laragen). Plasmids for overexpression in *S. cerevisiae* were constructed similarly except that the genes were PCRed from the cDNA of *S. araneosa*. All primers used in this study are listed in a separate file (Supplementary Table-sequences of oligonucleotides used in the study).

**Heterologous reconstitution of genes involved in sordaricin biosynthesis**. Fungal transformation was performed as follows. *A. nidulans* ΔEMΔST[31] was grown on oatmeal agar (Sigma) supplemented with uracil (5 mM), uridine (10 mM), riboflavin (0.125 μg/mL), and pyridoxine (0.5 μg/mL) at 28 °C for three days. Spores were filtered through a 40 μm cell strainer (Fisher), transferred to 45 mL CD media (1% glucose, 5% nitrate salt mix (0.12 g/mL NaNO₃, 0.104 g/mL KCl, 0.104 g/mL MgSO₄ · 7H₂O, 0.304 g/mL KH₂PO₄), 0.1% trace element mix (0.022 g/mL ZnSO₄ · 7H₂O, 0.011 g/mL H₃BO₃, 0.005 g/mL MnCl₂·4H₂O, 0.0016 g/mL FeSO₄ · 7H₂O, 0.0016 g/mL CoCl₂ · 5H₂O, 0.0016 g/mL CuSO₄ · 5H₂O, 0.0011 g/mL (NH₄)₆Mo₇O₂₄ · 4H₂O, pH 6.5)) supplemented with uracil (5 mM), uridine (10 mM), riboflavin (0.125 μg/mL), and pyridoxine (0.5 μg/mL) and shaken at 28 °C and 250 rpm for 12–16 h. This culture was centrifuged to collect the fungal body which was subsequently washed with 10 mL osmotic media (1.2 M MgSO₄, 10 mM sodium phosphate, pH 5.8). The resulting fungal body was resuspended in 10 mL osmotic media supplemented with 30 mg of Lysing enzyme from *Trichoderma harzianum* (Sigma) and 20 mg of Yatalase (Takara Bio) and shaken at 28 °C and 80 rpm for 5 h. The suspension was then poured into a centrifuge tube and 10 mL trapping buffer (0.6 M sorbitol, 0.1 M Tris, pH 7.0) was slowly added to the suspension so that the layers of fungal suspension and trapping buffer remained separated. The tube was then centrifuged at 4300 × *g* at 4 °C for 20 min. Trapped protoplast (greenish white layer at the interface of the fungal suspension and trapping buffer layers) was washed with 10 mL STC buffer (1.2 M sorbitol, 10 mM CaCl₂, 10 mM Tris, pH 7.5) and then resuspended in 1.2 mL STC buffer on ice. Plasmids (typically 150–500 ng) to be transformed were added to a 60 μL aliquot of the protoplast. The mixture was then incubated on ice for 1 h before addition of 600 μL PEG solution (60% PEG 4000, 50 mM CaCl₂, 50 mM Tris, pH 7.5) and further incubated at room temperature for 20 min. After the final incubation, the mixture was plated on CD-sorbitol agar (previously mentioned CD media supplemented with 1.2 M sorbitol and 2% agar) plates which were then incubated at 37 °C for 2–3 days until colonies appeared.

The resulting transformants were grown on CDST-Agar medium (2% starch, 2% casamino acid, 5% nitrate salt mix, 0.1% trace element mix, 2% agar) for 3–4 days at 28 °C. Two agar plugs (diameter ~ 1 cm) were cut from the plates and extracted with 700 μL acetone by vigorously vortexing for 20 min. The insoluble material was pelleted by centrifugation and the supernatant was dried in vacuum. The resulting residues were reconstituted with 100 μL HPLC grade methanol and centrifuged again to remove insoluble material. The supernatant was then directly analyzed by an Agilent 1260 Infinity II LC equipped with an InfinityLab Poroshell 120 EC-C18 column (2.7 μm, 3.0 × 50 mm) and a 6545 QTOF high resolution mass spectrometer (UCLA Molecular Instrumentation Center). The solvent program began with 1% acetonitrile for 2 min and then linearly increased to 90% acetonitrile over 9 min. The LC-MS data was acquired with MassHunter Workstation 10.0 (Agilent) and analyzed with MassHunter Qualitative Analysis 10.0 (Agilent).

For reconstitution in *S. cerevisiae* RC01, SdnA-B-H and SdnC-E-F were cloned into plasmids XW55 and XW06 respectively[56]. The two plasmids were subsequently transformed into *S. cerevisiae* RC01 with Frozen-EZ Yeast Transformation II kit (Zymo research). A 3 mL YPD culture (2% glucose, 1% yeast

extract, and 2% peptone) was inoculated with a single transformant colony and incubated in a 28 °C shaker, 250 rpm for 72 h. A 1 mL aliquot of this culture was then removed and centrifuged at $17,000 \times g$ for 5 min to separate the media and the cells which were then extracted separately with ethyl acetate and acetone, respectively. The organics were combined and dried under vacuum. The resulting residues were reconstituted with 100 μL HPLC grade methanol and analyzed similarly with *A. nidulans* reconstitution.

**Compound isolation and characterization.** Typically, 50 mL CDST agar plates of 2 L culture inoculated with *A. nidulans* transformants were placed in a 28 °C incubator for 3–4 days. The agar plates were cut into pieces and soaked in 2 L acetone for 24 h. Agar was removed by filtration and extracted again with acetone. The two extractions were combined and evaporated to dryness by a rotary evaporator. The residues were extracted with acetone and ethyl acetate. The crude extracts were separated by silica flash chromatography with a CombiFlash® system and a gradient of hexane and ethyl acetate. The targeted compounds were further purified by an UltiMate™ 3000 Semi-Preparative HPLC (ThermoFisher) with an Eclipse XDB-C18 column (5 μm, 9.4 × 250 mm, Agilent) and a gradient of water and acetonitrile (both mobile phases contain 0.1% formic acid). The gradient started with 30% acetonitrile and then linearly increased to 95% acetonitrile in 35 min. Purified compounds were dried in vacuum and analyzed by the Agilent UHPLC-HRMS as stated above and a Bruker AV500 NMR spectrometer with a 5 mm dual cryoprobe (500 MHz, UCLA Molecular Instrumentation Center). NMR data were collected with Topspin 3.5pl4 (Bruker) and analyzed with Topspin 4.1.1 (Bruker). Specific rotation was measured with an Autopol III Automatic Polarimeter equipped with a 50 mm polarimeter cell (Rudolph Research Analytical). The yield and spectroscopic data of each compound are listed in the Supplementary Notes.

**Heterologous biotransformation.** *A. nidulans* transformed with the gene of interest was grown on CDST-Agar fed with 200 μM of substrates for 3 days at 28 °C. Agar plugs from these plates were then extracted and analyzed as stated above. *A. nidulans* transformed with empty plasmids served as a control.

For biotransformation of compound **7** by SdnF, *A. nidulans* transformed with the SdnF was grown on a 5 mL CDST-Agar plate for 3 days. Compound **7** was dissolved in $CH_2Cl_2$ (2.5 mM final) and 10 μL of this solution was dropped directly onto the fungal mycelia. The plate was incubated at 28 °C for 24 h and an agar plug containing the fed compound was analyzed as stated above.

**General synthetic procedures for compounds 4, 7, and 10.** Compounds **4**, **7**, and **10** were synthesized from their corresponding reduced shunt products **3**, **5**, and **6** respectively by selective oxidation of the allylic alcohol with activated $MnO_2$[34]. Briefly, 20 mM alcoholic compounds were dissolved in $CH_2Cl_2$. Twenty molar equivalents of activated $MnO_2$ (Millipore) were added and the reaction mixture was vigorously stirred at room temperature for 2 h. $MnO_2$ was removed by centrifugation and the supernatant containing the desired product was generally used for downstream studies without the need of further purification.

**In vitro non-enzymatic IMDA cyclization of compounds 7 and 10.** Compound **7** or **10** was dissolved in a buffer of 50 mM HEPES, pH 7.4 with 5% DMSO (final concentration 100 uM). The reaction mixture was maintained at 25 °C with a MyBlock Mini Dry Bath (Benchmark Scientific) throughout the experiment. Aliquots (10 μL) of the reaction were taken at indicated time and immediately mixed with an equal volume of ice cold acetonitrile. The mixture was then directly analyzed by the previously mentioned UHPLC-QTOF. The solvent program was an isocratic gradient of 80% acetonitrile for 2.5 min. Decrease at $A_{305\ nm}$ (for compound **7**) or $A_{286\ nm}$ (for compound **10**) was used to monitor the IMDA cyclization. The remaining reactant was quantified by a standard curve of compound **7** or **10**. Concentration of the remaining reactant verse reaction time was fitted with first order kinetics to obtain the rate constant $k_{uncat}$ (GraphPad Prism 8).

**DFT calculation.** Initial conformational searches were conducted on all reported structures using xtb and CREST[57,58]. The output geometries were recalculated with the density function and basis set ωB97X-D/def2-SVP as implemented in Gaussian 16 Rev. A.03 (sse4)[59–62]. This functional was chosen for its ability to reproduce CCSD geometry calculations of asynchronous Diels–Alder reactions as well as its general applicability for accurately calculating reaction barriers[63]. Following Head-Gordon's suggested basis set for energetics[64], we computed single point energies at the ωB97X-D/def2-QZVPP level of theory with the CPCM implicit solvent model for water[65,66]. All reported energies are quasi-harmonic corrected for entropy and enthalpy[67,68]. Transition states were located by Berny optimization. The output geometry was subjected to a constrained conformational search to ensure that all substituents were in their lowest energy conformation. Then, the structure was re-optimized at the ωB97X-D/def2-SVP level of theory. Frequency calculations were conducted to verify whether each structure was indeed a transition state. Intrinsic reaction coordinate calculations were conducted to further verify the connectivity of the transition states. We have included one such calculation in the supporting information for TS-2: the calculation indicates that TS-2 connects reactant **10** to product **11** (Supplementary Fig. 82). The molecular coordinates of all calculated structures are listed in a separate Supplementary file (Supplementary Data-molecular coordinates of calculated structures).

**Cloning, expression, and purification of SdnG.** SdnG was PCR amplified from cDNA of *S. araneosa* and subcloned into a pET28a plasmid with a N-(His)$_6$-SUMO tag via NEBuilder® HiFi DNA Assembly Master Mix (NEB) (Supplementary Table 2). The resulting plasmid harboring N-(His)$_6$-SUMO-SdnG was transformed into chemically competent *E. coli* Rosetta 2 (Millipore). The transformant was used to inoculate 1 L LB medium with 50 μg/mL kanamycin and 34 μg/mL chloramphenicol. The culture was shaken at 37 °C and 220 rpm until $OD_{600}$ reached ~0.6–1 and cooled to 16 °C. IPTG was added to the culture to a final concentration of 100 μM and the culture was shaken at 16 °C for 16–20 h. After expression was completed, cells were harvested by centrifugation, flash-frozen in liquid nitrogen, and stored at −80 °C.

Cell pellet from 1 L culture (~5 g) was resuspended in cell lysis buffer (50 mM sodium phosphate, 500 mM sodium chloride, 10% glycerol, 25 mM imidazole, pH 8.0) by vortexing. All subsequent purification procedures were performed at 4 °C unless otherwise stated. Cells were lysed by sonication on ice (2 s on, 5 s off cycle for 24 min, 50% maximum amplitude) and then centrifuged for 30 min at 4 °C and $21,000 \times g$ to pellet cell debris. The supernatant was mixed with 1 mL Ni-NTA resin (Thermofisher, pre-equilibrated with 5 column volumes of cell lysis buffer) and agitated mildly for 1 h. The mixture was then loaded onto a gravity column. The resin was washed with at least 20 column volumes of cell lysis buffer. The N-(His)$_6$-SUMO-SdnG was eluted from the column with 10 column volumes of elution buffer (50 mM sodium phosphate, 500 mM sodium chloride, 10% glycerol, 250 mM imidazole, pH 8.0). Ulp1 protease (~40 μg) was then added to the eluted protein and the mixture was incubated on ice overnight to cleave the N-(His)$_6$-SUMO tag. The cleaved tag and Ulp1 protease which contains a N-(His)$_6$ tag were separated from SdnG by running the mixture through a fresh Ni-NTA column. SdnG, now (His)$_6$ tag free, passed through the column while the SUMO tag and Ulp1 protease remained bound. Purified SdnG was then concentrated by an Amicon spin concentrator (Millipore), aliquoted, flash-frozen in liquid nitrogen, and stored at −80 °C. The purified SdnG was analyzed by 12% SDS-PAGE and protein concentration was estimated from its predicted extinction coefficient ($\varepsilon_{280\ nm} = 8940\ cm^{-1}\ M^{-1}$)[69].

**In vitro assay of SdnG.** SdnG (0.18 μM for compound **10** and 10 μM for compound **7**) was preincubated in 95 μL of 50 mM HEPES, pH 7.4 buffer for 5 min. This solution was maintained at 25 °C with a MyBlock Mini Dry Bath (Benchmark Scientific) throughout the experiment. Reaction was initiated by adding 5 μL DMSO solution of compounds **10** or **7**. The final concentration of the substrate is 20–175 μM for compound **10** and 100 μM for compound **7**. After 1 min, 10 μL of the reaction were taken and immediately mixed with an equal volume of ice cold acetonitrile. The mixture was then directly analyzed by UHPLC-QTOF same as the non-enzymatic cyclization reactions. The measured rate was subtracted with the rate of non-enzymatic cyclization to obtain the true enzymatic reaction rate. The concentration of compound **10** verse rate was fitted with Michaelis-Menton equation to deduce steady-state kinetics parameters (GraphPad Prism 8).

**Reporting summary.** Further information on research design is available in the Nature Research Reporting Summary linked to this article.

## Data availability

All data supporting the findings of this study are available within the article and its Supplementary Information files. All unique biological materials, such as plasmids, generated in the study are available from the authors. The DNA sequence of *sdn* cluster (Accession: LC079035.1) and genomes of *Rosellinia necatrix* (Accession: ASM2120987v1) and *Xylaria hypoxylon* (Accession: GCA_902806585.1) are available from NCBI. The protein sequence of SdnG (Accession: A0A1B4XBH4) is available from Uniprot. Source data are provided with this paper.

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

## Acknowledgements

This work was supported by the NIH grant 1R01AI141481 to Y.T. and K.N.H. C.S.J. is supported by generous funding through the Saul Winstein Fellowship. We thank Wenyu Han for his help with cloning and Prof. Christopher T. Walsh for the critical reading and comments of this manuscript.

## Author contributions

Z.S., C.S.J., M.O., K.N.H., and Y.T. developed the hypothesis and designed the study. Z.S. performed all in vivo and in vitro studies. C.S.J. performed all computational studies. Z.S., C.S.J., M.O., K.N.H., and Y.T. analyzed and discussed the results and prepared the manuscript.

## Competing interests

The authors declare no competing interests.
