## [Peer Review File · Nature Communications]

REVIEWER COMMENTS

Reviewer #1 (Remarks to the Author):

This is an excellent and thorough piece of work, adding to the rapidly growing canon of known pericyclase enzymes - notably provided by fungi. Here a rare DA step in a terpene pathway is dissected that has several novel and highly interesting facets. This is the first DAase catalysing a reaction with a cyclic diene. The flow of chemical logic is extremely elegant and will be of high interest to synthetic as well as biosynthetic chemists. The use here of 'total mycosynthesis' represents both a perfect way of delineating the pathway, but also an interesting test-bed and platform for future pathway engineering. The combination of experimental and theoretical aspects to this study makes the conclusions highly convincing. The data in support is of very high quality. Overall I have only minor comments:

1. The case of the solanopyrones could be mentioned as being related - there is an interesting redox cycle here from alcohol to activated aldehyde and then back to alcohol after the DA step.
2. Position numbers are normally C-8, O-15 H-4/C-5 etc
3. line 226 - ¹H and ¹³C the numbers should be superscript.
4. The calls to the 'extended' data are annoying. Its 2022 - why can't this be included in the main paper? Its not like the journal has to save on ink and paper....

Reviewer #2 (Remarks to the Author):

The manuscript is a complete work to discover a terpene Biosynthetic Pathway, activating by norbornene-diels-alderase.

The authors not only isolate the enzymes necessary to carry out the study but also synthesize the intermediates and perform DFT studies to confirm their initial hypotheses.

The results are robust and constitute an important contribution to the field of biosynthesis of natural products.

It is also important to note the quality of the supplementary material. However, authors are advised to include enlargements of the two-dimensional spectra, making it easier for readers to use them.

The high quality of the manuscript allows to recommend it be published in its current state.

Reviewer #3 (Remarks to the Author):

This manuscript describes comprehensive elucidation of the core sordaricin biosynthetic pathway, with particularly intriguing discovery of a novel pericyclase catalyzing the heretofore unknown Diels-Alder cyclization between a cyclopentadiene and dienophile to form a norborene ring structure. As prominently noted here, this is a commonly utilized and studied cycloaddition in synthetic chemistry, but had previously not been seen in a biosynthetic pathway. While the relevant diterpene cyclase SdnA had been previously identified, along with an extended biosynthetic gene cluster (BGC) associated with production of the extensively more elaborated hypoxysordarin, here the subsequently acting cytochromes P450 SdnB, E, F and H, as well as novel pericyclase SdnG were identified. This was accomplished by discovery of more compact BGCs associated with production of the simpler sordarins, which enabled more focused studies of the above-mentioned enzymes. Notably, SdnB was found to play multiple roles, both serving to di-hydroxylate neighboring carbons in the tricyclic cycloaraneosene diterpene produced by SdnA, but also latter scission between these hydroxylated carbons, with this oxidative cleavage generating a dialdehyde derivative. Interestingly, this only occurs following an intervening dehydrogenation reaction catalyzed by SdnH. Although a ring cleavage product is observed in the absence of SdnH, data is shown indicating that such cleavage is more readily accomplished following formation of the cyclopentadiene by SdnH. This oxidative ring cleavage is followed by further oxidation of the aldehyde substituent of the cyclopentadiene moiety to a carboxylate, which enables the Diels-Alder cycloaddition to form the norborene ring structure. Although this can occur spontaneously, with subsequent hydroxylation catalyzed by SdnE then producing sordaricin, observation of significant amounts of shunt products led to further investigation of SdnG, which was found to act as a pericyclase increasing the Diels-Alder cycloaddition reaction (>500-fold). Intriguingly, SdnG is a novel sequence, representing a new class of pericyclases, discovery of which adds further significance to the reported studies. Indeed, given the noted concise oxidative strategy and novel pericyclase revealed here, in conjunction with the long-sought nature of the catalyzed Diels-Alder cyclization between a cyclopentadiene and dienophile to form a norborene, this study should be of reasonably wide-spread interest.

Minor corrections/clarifications:

Please ensure that “sordaricin” is correctly spelled throughout (“sodaricin” often observed).

Line 136 – Extended data fig. 3a does not demonstrate oxidative cleavage of 2 to 4, as only 3 is observed. Instead, it is a combination of that experiment and the one shown in extended data fig. 3d, where feeding of 4 is shown result in efficient conversion to 3. Thus, both should be cited here.

c.f. Line 146 & Fig. 2b – Please discuss how 6 is generated in the absence of SdnF.

Reviewer #4 (Remarks to the Author):

The Authors present an investigation concerning the discovery and characterization of a terpene biosynthetic pathway featuring a Norbornene-forming Diels-Alderase.

I cannot comment on the biological and biochemical relevance of the manuscript since my expertise is in computational organic chemistry and NMR spectroscopy. Therefore my review below is limited to the technical aspects of these two techniques.

Concerning the DFT calculations, these appears to be run at a rather high level of theory. The functional used, w-B97XD, is a modern functional which includes a treatment of dispersive interactions and long range corrections - necessary when dealing with molecules with pi-electrons and where van der Waals interactions are expected to play a significant role. The basis set is also sufficiently large. The level of theory used for the calculation is indeed adequate.

However the Authors do not comment on the method used to calculate the transition states. In Figure 3 they report transition states, how these stationary points were calculated? Did the Authors check that the TS was actually connecting reactants and products by following the reaction path?

Concerning the NMR part, the intermediates have been thoroughly characterized by ¹H, ¹³C, COSY, HSQC and HMBC NMR (as well as HR mass) with a high resolution 500 MHz instrument. When necessary, a NOESY spectrum is also included (please check the spelling in Supplementary Material, it is reported as NOSEY instead of NOESY).

In short, the computational and NMR parts are very well executed, except for some minor issues mentioned above.

We thank the reviewers for their comments and suggestions. We have provided a list of replies and improvements in the following table. The page and line number mentioned below correspond to the pdf file with track change.

Reviewer 1	
1. The case of the solanopyrones could be mentioned as being related - there is an interesting redox cycle here from alcohol to activated aldehyde and then back to alcohol after the DA step.	i) We mentioned solanopyrone as a related case to the oxidative diene activation of sordaricin biosynthesis in the first submission. In the revised manuscript, we expanded the corresponding sentence (as showing below) and added an additional citation to further elaborate. Page18, line 345-348: “Such redox activation of diene/dienophile is analogous to the role played by the flavin-dependent enzyme Sol5 catalyzing a decalin-forming IMDA in solanapyrone biosynthesis where Sol5 is required to activate the dienophile by oxidizing an electron donating hydroxymethyl group conjugated to the dienophile to an electron withdrawing aldehyde^{45,46}”.
2. Position numbers are normally C-8, O-15 H-4/C-5 etc	ii) A hyphen was inserted in between atom labels and atom numbers throughout the revised manuscript.
3. line 226 - 1H and 13C the numbers should be superscript.	iii) This was corrected throughout the revised manuscript.
4. The calls to the 'extended' data are annoying. Its 2022 - why can't this be included in the main paper? Its not like the journal has to save on ink and paper....	iv) We moved Extended Data Figs 3, 4, and 5 to the main text where they are now Figs 3, 5, and 6. Extended Data Figs 1 and 2 were moved to the Supplementary Information as Supplementary Fig. 1 and 3.
Reviewer 2	
Authors are advised to include enlargements of the two-dimensional spectra, making it easier for readers to use them.	We attempted to enlarge the spectra but the figure width is already at the limit of the page width. However, the pdf file we resubmitted, when zoomed in, provides reasonable resolution for the 2D spectra. We hope this will satisfy the readers.
Reviewer 3	
1. Please ensure that “sordaricin” is correctly spelled throughout (“sodaricin” often observed).	i) This has been corrected throughout the revised manuscript.

2. Line 136 – Extended data fig. 3a does not demonstrate oxidative cleavage of 2 to 4, as only 3 is observed. Instead, it is a combination of that experiment and the one shown in extended data fig. 3d, where feeding of 4 is shown result in efficient conversion to 3. Thus, both should be cited here.	ii) We added a trace showing efficient conversion of 4 to 3 by A. nidulans to Extended data fig. 3a (now Fig. 3a, relabeled following the suggestion of another reviewer). We also revised the corresponding paragraph as follows to clarify this point. Page 7, line 132-151: “We proposed that 3 is likely formed via oxidative cleavage of the C-8,C-9 diol in 2 by SdnB (Fig. 2b), which was confirmed through direct feeding of 2 to A. nidulans expressing only SdnB (Fig. 3a). Compound 2 is stable under the same feeding conditions when an empty plasmid control was used. Since the conversion of 2 to 3 is a net redox-neutral process, the product of SdnB oxidation is likely dialdehyde 4 instead of 3. In the absence of downstream enzymes, the C-7 acrolein moiety in 4 can be reduced by endogenous reductases in A. nidulans to afford alcohol 3 as a cellular detoxification mechanism to remove the unsaturated aldehyde^{33,34}. We chemically synthesized 4 by selectively oxidizing the allylic alcohol in 3 to the corresponding unsaturated aldehyde via activated MnO₂³⁵ (Supplementary Notes, Supplementary Table 7, Supplementary Figs. 2, 23-27). Consistent with our hypothesis, 4 was readily converted to 3 when fed to A. nidulans expressing only empty plasmids (Fig. 3a). Overall, our results suggest that SdnB both acts as a canonical monooxygenase and also a “thwarted oxygenase”, an oxygenase that consumes oxygen to generate strong enzymatic oxidant but does not result in formal incorporation of oxygen atom into the product³² (Supplementary Fig. 3). The diol cleavage activity of SdnB enables rotation of the C-6-C-7 bond and thereby “freed” the C-7-C-17 double bond which is the proposed dienophile for the IMDA reaction (Fig. 2b).”
3. Line 146 & Fig. 2b – Please discuss how 6 is generated in the absence of SdnF.	iii) We believe in the absence of SdnF, 6 can be generated from 5 either by auto-oxidation or host oxidases. As shown in Extended Data Fig. 4b (now Fig. 5b, relabeled following the suggestion of another reviewer), feeding of 5 to A. nidulans expressing only empty plasmid resulted in formation of 6. We

	modified the corresponding sentence as follows to incorporate this discussion. Page 10, line 167-170: “Similar to the formation of 3 from 4, 5 is likely a redox shunt product derived from cyclopentadiene-dialdehyde 7, which should be the product of sequential actions of SdnB and SdnH starting from cycloaraneosene (and 2). Auto-oxidation or host oxidases may subsequently convert 5 to 6.” Page 13, line 235-240: “To assay the oxidation activities of SdnF separately from cyclization of 10, 5 was supplemented to the SdnF expression strain and a control strain expressing only empty plasmids. While the control strain is able to convert ~30% of 5 to the carboxylate 6 likely via auto-oxidation or host oxidases, the SdnF expression strain led to near complete conversion of 5 to 6 (Fig. 5b), demonstrating that SdnF is able to selectively oxidize the C-8 aldehyde to carboxylate.”
Reviewer 4	
1. The Authors do not comment on the method used to calculate the transition states. In Figure 3 they report transition states, how these stationary points were calculated? Did the Authors check that the TS was actually connecting reactants and products by following the reaction path?	i) The stationary points corresponding to the transition states were located by guessing an initial geometry (based upon established TS geometries for Diels–Alder reactions of cyclopentadiene and dienophiles) and optimizing with the Gaussian keywords [opt=(calcfc,ts,noeigen)]. Force constants (calcfc) were calculated as an initial guess for the Berny optimization and the eigen test (noeigen) was forgone in an effort to force the calculation to continue to try to find the TS even if the curvature in the Berny optimization was unfavorable. (noeigen is only recommended if you have substantial computing power) Upon locating the transition state, a constrained conformational search was conducted using CREST in order to verify that all substituents were in their lowest energy conformation and then the output geometry was re-optimized at the DFT level of theory. Frequency calculations on the transition state structures indicate that the imaginary frequency leads from reactants to products.

	We have confirmed that all transition states do connect from reactants to products and we have added in an example intrinsic reaction coordinate calculation for TS-2 to the supplemental information (Supplementary Fig. 82). We added the following paragraph to the DFT Method section to describe the Transition State calculations. Page 30, line 602-608: “Transition states were located by Berny optimization. The output geometry was subjected to a constrained conformational search to insure that all substituents were in their lowest energy conformation. Then, the structure was re-optimized at the ωB97X-D/def2-SVP level of theory. Frequency calculations were conducted to verify whether each structure was indeed a transition state. Intrinsic reaction coordinate calculations were conducted to further verify the connectivity of the transition states. We have included one such calculation in the supporting information for TS-2: the calculation indicates that TS-2 connects reactant 10 to product 11 (Supplementary Fig. 82).”
2. Please check the spelling in Supplementary Material, it is reported as NOSEY instead of NOESY	ii) This has been corrected.

REVIEWERS' COMMENTS

Reviewer #1 (Remarks to the Author):

Publish as is

Reviewer #3 (Remarks to the Author):

This manuscript has been suitably revised in response to my previous comments and I have no further issues.

Reviewer #4 (Remarks to the Author):

The Authors have revised the manuscript in a satisfactory way. It can be now published in its present form.